# Application of Feedforward and Recurrent Neural Networks for Fusion of Data from Radar and Depth Sensors Applied for Healthcare-Oriented Characterisation of Persons’ Gait

**DOI:** 10.3390/s23031457

**Published:** 2023-01-28

**Authors:** Paweł Mazurek

**Affiliations:** Warsaw University of Technology, Faculty of Electronics and Information Technology, Institute of Radioelectronics and Multimedia Technology, ul. Nowowiejska 15/19, 00-665 Warsaw, Poland; pawel.mazurek@pw.edu.pl

**Keywords:** measurement data fusion, neural networks, impulse-radar sensor, depth sensor, healthcare

## Abstract

In this paper, the useability of feedforward and recurrent neural networks for fusion of data from impulse-radar sensors and depth sensors, in the context of healthcare-oriented monitoring of elderly persons, is investigated. Two methods of data fusion are considered, viz., one based on a multilayer perceptron and one based on a nonlinear autoregressive network with exogenous inputs. These two methods are compared with a reference method with respect to their capacity for decreasing the uncertainty of estimation of a monitored person’s position and uncertainty of estimation of several parameters enabling medical personnel to make useful inferences on the health condition of that person, viz., the number of turns made during walking, the travelled distance, and the mean walking speed. Both artificial neural networks were trained on the synthetic data. The numerical experiments show the superiority of the method based on a nonlinear autoregressive network with exogenous inputs. This may be explained by the fact that for this type of network, the prediction of the person’s position at each time instant is based on the position of that person at the previous time instants.

## 1. Introduction

The life expectancy at birth, estimated for the global population in 2019, was, ca., 73 years; it has been rising during the last decades and is predicted to reach 77 years by the second half of the twenty-first century. At the same time the global fertility rate (i.e., the number of live births per woman over a lifetime) is decreasing [1]. As a consequence, the global population is ageing—it is expected that the share of European and North American population aged at least 65 years will reach 25% in 2050. Hence, there is a growing importance of research on new technologies that could be employed in monitoring systems supporting care services for elderly persons. The falls of elderly persons belong to the most frequent reasons for their admission and long-term stay in hospitals [2], and therefore the monitoring systems developed for the sake of those persons should enable prediction and prevention of dangerous events, such as person’s fall and harmful long lie after the fall, and also detection of those events [3,4]. The relevance of features related to gait analysis in monitoring of elderly persons, and in particular in fall prevention, has been emphasised in several papers [5,6,7,8].

Elderly persons are often reluctant to use the existing monitoring techniques because they may infringe on the privacy of those persons, or require the constant wearing of some additional devices [9,10]. As a result, two relatively new non-invasive and non-intrusive monitoring techniques are attracting growing attention of the researchers, viz., techniques based on depths sensors [11,12,13,14,15,16,17,18,19,20] and radar sensors [21,22,23,24,25,26,27,28,29,30,31,32].

A system based on the impulse-radar sensors and depth sensors may not only be used for localisation of a monitored person in their apartment, but also for estimation of several healthcare-related parameters, enabling medical personnel to make useful inferences on the health condition of that person [33,34]. These parameters include, but are not limited to: the estimates of the time spent in motion and travelled distance (both helpful in assessing the level of physical activity), the number of turns made during motion (considered an indicator of mental health condition [35]) and the mean walking speed (considered highly informative with respect to the overall health status [36]).

It has been recently shown by the author and his collaborators that the fusion of measurement data from the impulse-radar sensors and depth sensors may enable reliable estimation of the monitored person’s position [37], and that a multi-layer perceptron, trained on synthetic data, may be effectively used for that purpose [38]. In the case of the monitoring of elderly persons, training of neural networks may be a particularly challenging task: involving elderly persons in the experimentation aimed at the acquisition of the reference data could be problematic both for medical and ethical reasons. The use of synthetic data for training of these networks may solve this problem.

In this paper, the idea of applying artificial neural networks, trained on synthetic data, for fusion of the measurement data from the impulse-radar sensors and depth sensors, is further explored. The multilayer perceptron treats all the measurement data independently and does not take into account the movement history, i.e., the dependence of the current position and direction of the movement on the past position and direction of the movement. This characteristic of a movement may be exploited in the process of data fusion by means of a recurrent neural network, which takes into account the previous values of its output for predicting the current value of its output.

The novelty of the research results presented in this paper consists in application of a new method, based on a nonlinear autoregressive network with exogenous inputs, for the fusion of data from impulse-radar sensors and depth sensors, as well as a systematic comparison of that method with a method based on a multilayer perceptron, and one reference method within a comprehensive programme of experimentation, based on the real-world data, involving:The localisation of a person walking around the monitored area according to various predefined movement scenarios.The estimation of several parameters, carrying information important for medical experts, on the basis of the estimated movement trajectories.

This programme included, in particular, experiments aimed at the investigation of the influence of the obstacles, occluding a monitored person, and the walking speed of that person on the accuracy of the estimation of the parameters.

## 2. Compared Methods of Data Fusion

### 2.1. Method Based on Nonlinear Autoregressive Network with Exogenous Inputs

This method (called *NARX method* hereinafter) is based on a nonlinear autoregressive network with exogenous inputs, with the following structure (Figure 1a):Four input neurons—the input data represent two pairs of the *x*-*y* coordinates acquired by means of the impulse-radar sensor and depth sensor;Six neurons in a single hidden layer;Two output neurons—the output data represent one pair of fused *x*-*y* coordinates; the output values, delayed by two time instants, are fed back to the hidden layer;

The implementation of the neural network has been based on the procedures available in the MATLAB Deep Learning Toolbox [39].

The artificial neural network was trained on the synthetic data generated according to the methodology described in detail in [38]. The synthetic data representative of the measurement data acquired by means of the impulse-radar sensors were corrupted with zero-mean red noise and smoothed using the moving-average filter. The generation of the synthetic data representative of the measurement data acquired by means of the depth sensor involved the modelling of a silhouette of a moving person by means of an ellipse following a sine-shaped trajectory oscillating around the reference trajectory; generation of those data took into account the fact that the depth sensors can “see” only one side of the body.

Four different reference trajectories, presented in Figure 2, were used for generation of the synthetic data. For each trajectory, forty realisations of the data were generated for each type of sensor, but in the case of a depth sensor, twenty of those realisations were fragmented to represent the corruption of the measurement data due to the occlusion. The data acquisition rate for the depth sensor and for the impulse-radar sensors was set to 10 Hz. All the generated trajectories are presented in Figure 3.

### 2.2. Method Based on Multilayer Perceptron

This method (called *MLP method* hereinafter), introduced by the author in [39], is based on a multilayer perceptron with four input neurons, eight neurons in a single hidden layer, and two output neurons (Figure 1b).

### 2.3. Method Based on Kalman Filter

This method (called *KF method* hereinafter), described in [37], has been chosen as the reference method because it yielded very good results in an extensive experiment based on real-world data; moreover, the KF method (as the NARX method and the MLP method) is a method of data-point fusion, i.e., it is based on the assumption that the fusion of the data acquired by means of the impulse-radar sensors and the data acquired by means of the depth sensor is performed whenever new data points are available.

This method is derived from linear transition equation modelling the movement of a monitored person walking with nearly constant velocity, viz., with a small random acceleration between two consecutive time instants [40]:(1)$$x_{n}=F_{n}x_{n-1}+\Gamma_{n}\alpha_{n}$$
where:

$x_{n}={\left[x_{n}\;x_{n}^{\left(1\right)}\;y_{n}\;y_{n}^{\left(1\right)}\right]}^{T}$ is a state vector representative of the two-dimensional coordinates $\left(x_{n},\;y_{n}\right)$, corresponding to a time instant $t_{n}$, and the velocities along these dimensions $\left(x_{n}^{\left(1\right)},y_{n}^{\left(1\right)}\right)$;$F_{n}$ and $\Gamma_{n}$ are the matrices of the form:(2)$$F_{n}=\left[\begin{array}{cccc} 1 & \Delta_{n} & 0 & 0\\ 0 & 1 & 0 & 0\\ 0 & 0 & 1 & \Delta_{n}\\ 0 & 0 & 0 & 1 \end{array}\right]\;\mathrm{and}\;\Gamma_{n}=\left[\begin{array}{cc} \Delta_{n}^{2}/2 & 0\\ \Delta_{n} & 0\\ 0 & \Delta_{n}^{2}/2\\ 0 & \Delta_{n} \end{array}\right]$$
with $\Delta_{n}$ being the time interval between the time instants $t_{n-1}$ and $t_{n}$;$\alpha_{n}={\left[\alpha_{x,n}\;\alpha_{y,n}\right]}^{T}$ is a vector modelling the acceleration, whose elements are assumed to be the realisations of a zero-mean bivariate normal distribution with a known covariance matrix $\Sigma_{\alpha }=\mathrm{diag}\left\{\sigma_{\alpha ,x}^{2},\sigma_{\alpha ,y}^{2}\right\}$.

The output equation, complementing the transition Equation (1), is modelling the relationships between the vector of observations ${\tilde{z}}_{n}$ and the state vector $x_{n}$:(3)$${\tilde{z}}_{n}=Hx_{n}+\eta_{n}$$
where:

H is an observation matrix of the form:(4)$$H=\left[\begin{array}{c} \begin{array}{cccc} 1 & 0 & 0 & 0 \end{array}\\ \begin{array}{cccc} 0 & 0 & 1 & 0 \end{array} \end{array}\right]$$$\eta_{n}={\left[\eta_{x,n}\;\eta_{y,n}\right]}^{T}$ is a vector representative of the observation noise corrupting the data, whose elements are assumed to be realisations of a zero-mean bivariate normal distribution with a known covariance matrix $\Sigma_{\eta ,n}$.

Under the above-formulated assumptions, the radar data and the depth data can be fused by means of a Kalman filter (*cf*. the references [41,42,43]) performing, for each time instant, the following sequence of operations [40,44]:

1.Determination of the pre-estimate ${\hat{x}}_{n}^{\mathrm{pre}}$ of the state vector and the pre-estimate ${\hat{P}}_{k}^{\mathrm{pre}}$ of its covariance matrix:

(5)$${\hat{x}}_{n}^{\mathrm{pre}}=F_{n}{\hat{x}}_{n-1}$$(6)$${\hat{P}}_{n}^{\mathrm{pre}}=F_{n}{\hat{P}}_{n-1}F_{n}^{T}+Q_{n}$$
where:(7)$$Q_{n}=\left[\begin{array}{cccc} \frac{1}{4}\Delta_{n}^{4}\sigma_{\alpha ,x}^{2} & \frac{1}{2}\Delta_{n}^{3}\sigma_{\alpha ,x}^{2} & 0 & 0\\ \frac{1}{2}\Delta_{n}^{3}\sigma_{\alpha ,x}^{2} & \Delta_{n}^{2}\sigma_{\alpha ,x}^{2} & 0 & 0\\ 0 & 0 & \frac{1}{4}\Delta_{n}^{4}\sigma_{\alpha ,y}^{2} & \frac{1}{2}\Delta_{n}^{3}\sigma_{\alpha ,y}^{2}\\ 0 & 0 & \frac{1}{2}\Delta_{n}^{3}\sigma_{\alpha ,y}^{2} & \Delta_{n}^{2}\sigma_{\alpha ,y}^{2} \end{array}\right]$$

2.Calculation of the estimate ${\hat{z}}_{n}$
of the observation vector:



(8)
$${\hat{z}}_{n}=H{\hat{x}}_{n}^{\mathrm{pre}}$$



3.Calculation of the so-called *innovation vector*:
(9)$$\zeta_{n}={\tilde{z}}_{n}-{\hat{z}}_{n}$$
and the estimate of its covariance matrix:(10)$${\hat{\Sigma }}_{\zeta ,n}=H{\hat{P}}_{n}^{\mathrm{pre}}H^{T}+\Sigma_{\eta ,n}$$
for each vector of data ${\tilde{z}}_{n}$;

4.Calculation of the final estimate of the state vector:
(11)$${\hat{x}}_{n}={\hat{x}}_{n}^{\mathrm{pre}}+G_{n}\zeta_{n}$$
and the final estimate of its covariance matrix:

(12)$${\hat{P}}_{n}=\left(I_{N}-G_{n}H\right){\hat{P}}_{n}^{\mathrm{pre}}$$
where $G_{n}$ is the Kalman-gain matrix determined according to the formula:



(13)
$$G_{n}={\hat{P}}_{n}^{\mathrm{pre}}H^{T}{\hat{\Sigma }}_{\zeta ,n}^{-1}$$



It is worth noting that, in this method, the radar data and depth data are acquired asynchronously; therefore, an observation vector ${\tilde{z}}_{n}$ may be representative of either the radar data or the depth data, and is always associated with a corresponding known covariance matrix $\Sigma_{\eta ,n}$ of the noise corrupting those data. The final estimate ${\hat{x}}_{n}$ of the state vector is the result of data fusion in every time instant $t_{n}$.

## 3. Extraction of Healthcare-Related Parameters

### 3.1. Detection of Motion

Motion has been detected by comparing the distance travelled in a given period of time, $T_{m}$, with a given threshold, D. A binary sequence, indicating motion, has been determined according to the formula:(14)$$b_{m,n}\equiv \left\{ \begin{array}{l} 1\;\mathrm{if}\;\;d_{n}>\frac{t_{n}-t_{n_{0}}}{T_{m}}D\\ 0\;\mathrm{otherwise} \end{array}\right.\;\mathrm{for}\;n=1,\;\ldots ,\;N$$
where:(15)$$n_{0}\equiv \arg\;\inf_{\nu }\left\{\nu |\;t_{\nu }\geq t_{n}-T_{m},\;\nu =0,\;\ldots ,\;n-1\right\}$$
(16)$$d_{n}\equiv \sqrt{{\left(x_{n}-x_{n_{0}}\right)}^{2}+{\left(y_{n}-y_{n_{0}}\right)}^{2}}$$

In the above equations, $\left\{x_{n}\right\}$ and $\left\{y_{n}\right\}$ are the sequences of the coordinates of the monitored person’s position, being either acquired by means of the sensors or fused by means of the methods described in Section 2. The values of $T_{m}$ and D should prevent small deviations in the position from being considered as motion; $T_{m}=0.45\;s$ and D=0.1 m have been selected because these values yielded satisfactory results in the experiments.

To reduce the influence of the dispersion of the position estimates on the detection of motion, morphological closing [45] has been performed on the binary sequence $\left\{b_{m,n}\right\}$. Because the fused data sequences are acquired by means of sensors with different data acquisition rates, the length of the structuring element used for the morphological closing has been equal to the number of data points within the time interval $T_{m}$. The binary sequence obtained by means of the morphological closing, $\left\{{\bar{b}}_{m,n}\right\}$, has been used for the detection of the moments when a monitored person started walking, and the moments when that person stopped walking.

### 3.2. Estimation of Walking Direction and Moment of Turning

The walking direction has been estimated by computing the four-quadrant inverse tangent of two consecutive positions $\left(x_{n},y_{n}\right)$ and $\left(x_{n-1},y_{n-1}\right)$ of the walking person. It is assumed that the counter domain of the function is [−π,π] or [−180°,180°].

A turn can be detected by comparing the absolute value of the sum of changes in the walking direction in a given period of time, $T_{t}$, with a given threshold, Φ. A binary sequence, indicating turning, has been computed according to the formula:(17)$$b_{t,n}\equiv \left\{ \begin{array}{l} 1\;\mathrm{if}\;\Phi <\left|\sum_{\nu =n_{0}+2}^{n}\arctan\left(\frac{x_{\nu }-x_{\nu -1}}{y_{\nu }-y_{\nu -1}}\right)-\arctan\left(\frac{x_{\nu -1}-x_{\nu -2}}{y_{\nu -1}-y_{\nu -2}}\right)\right|<\pi \\ 0\;\mathrm{otherwise} \end{array}\right.\;\mathrm{for}\;n=1,\;\ldots ,\;N$$

In the above equations, $\left\{x_{n}\right\}$ and $\left\{y_{n}\right\}$ are the sequences of the coordinates of the monitored person’s position. The values of $T_{t}$ and Φ should be optimised to prevent small deviations in the position from being considered as turns; $T_{t}=1\;s$ and Φ≅1.05 rad (60°) have been selected because these values yielded satisfactory results in the experiments.

To reduce the influence of the dispersion of the position estimates on the detection of turns, morphological closing has been performed on the binary sequence $\left\{b_{t,n}\right\}$. Because the fused data sequences are acquired by means of sensors with different data acquisition rates, the length of the structuring element used for the morphological closing has been equal to the number of data points within the time interval $T_{t}$. The binary sequence obtained by means of the morphological closing, $\left\{{\bar{b}}_{t,n}\right\}$, has been used for the detection of the moments when a monitored person started turning, and the moments when that person stopped turning.

### 3.3. Estimation of Travelled Distance

The travelled distance has been calculated by summing up the distances between the consecutive locations of the walking person:(18)$$s=\sum_{n=2}^{N}\sqrt{{\left(x_{n}-x_{n-1}\right)}^{2}+{\left(y_{n}-y_{n-1}\right)}^{2}}$$

### 3.4. Estimation of Walking Speed

The estimation of the average walking speed has been calculated by dividing the estimate of the travelled distance, determined according to Equation (18), by the estimate of the time spent in motion, determined according to Equation (14).

## 4. Methodology of Experimentation

### 4.1. Acquisition of Measurement Data

In the experiments, the raw measurement data were acquired by means of:Two impulse-radar sensors based on Novelda NVA6201 (Novelda, Oslo, Norway) chip working in the frequency range 6.0–8.5 GHz [46], and having the data acquisition rate of 10 Hz;An infrared depth sensor being a part of the Microsoft Kinect V2 device (Microsoft, Redmond, WA, United States) [47], having the data acquisition rate of 30 Hz.

The experimentation programme comprised two parts (named EXP#1 and EXP#2) involving the monitoring of the movements of a person walking within the area presented in Figure 4a:

In experiment EXP#1, a person walked forth and back along a serpentine trajectory, among the obstacles occluding that person (see Figure 4b); 30 walks were performed and the walking speed was kept constant at v=0.5 m/s.

In experiment EXP#2, a person walked clockwise and counter-clockwise along a rectangle-shaped trajectory (see Figure 4c), at six different values of the walking speed v∈{0.5, 0.6, 0.7, 0.8, 0.9, 1.0} m/s. For each value of the walking speed, 20 walks were performed.

Thus, the whole programme of experimentation comprised the acquisition of R=150 sequences of *x*-*y* data. Throughout the experimentation, the constancy of the walking speed was assured by means of a metronome.

### 4.2. Criteria for Performance Evaluation

#### 4.2.1. Indicators of Uncertainty of Estimation of Monitored Person’s Position

For each sequence of the data representative of the person’s position on the *x*-*y* plane:(19)$$\left\{{\hat{x}}_{n,r}|n=1,\ldots ,N_{r}\right\}\;\mathrm{and}\;\left\{{\hat{y}}_{n,r}|n=1,\ldots ,N_{r}\right\}\;\mathrm{for}\;r=1,\;\ldots ,\;R$$
the corresponding sequence of the absolute errors of the position estimation has been computed by subtraction of the reference values:(20)$$\left\{\Delta {\hat{x}}_{n,r}={\hat{x}}_{n,r}-{\dot{x}}_{n,r}|n=1,\ldots ,N_{r}\right\}\;\mathrm{and}\;\left\{\Delta {\hat{y}}_{n,r}={\hat{y}}_{n,r}-{\dot{y}}_{n,r}|n=1,\ldots ,N_{r}\right\}\;\mathrm{for}\;r=1,\;\ldots ,\;R$$

Finally, the sequences of the absolute errors of the position estimation, viz. $\left\{\Delta {\hat{x}}_{n,r}\right\}$ and $\left\{\Delta {\hat{y}}_{n,r}\right\}$, have been used for determining the R sequences of the position errors $\left\{\Delta {\hat{d}}_{n,r}\right\}$:(21)$$\left\{\Delta {\hat{d}}_{n,r}=\sqrt{{\left(\Delta {\hat{x}}_{n,r}\right)}^{2}+{\left(\Delta {\hat{y}}_{n,r}\right)}^{2}}|n=1,\ldots ,N_{r}\right\}\;\mathrm{for}\;r=1,\;\ldots ,\;R$$

The assessment of the performance of the compared methods of data fusion has been based on the inspection of the empirical cumulative distribution functions F(ξ) [48] characterising the position errors:(22)$$F\left(\xi \right)\equiv \frac{1}{M}\sum_{n=1}^{M}I\left(\Delta {\hat{d}}_{m}\leq \xi \right)\;$$
where $\left\{\Delta {\hat{d}}_{m}|m=1,\ldots ,M\right\}$ is a sequence composed of all the position errors $\left\{\Delta {\hat{d}}_{n,r}\right\}$, and I( • ) is the function taking the value 1 if the condition inside the brackets is met, and the value is 0 otherwise. The following indicators of uncertainty of estimation have been used:

The area under the empirical cumulative distribution function F(ξ) in the interval ξ∈[0,1] m, denoted with $A_{\mathrm{ECDF}}$, taking the value from the interval [0,1];

The mean position error (MEAE);The maximum position error (MAXE);The standard deviation of the position errors (STDE).

#### 4.2.2. Indicators of Uncertainty of Estimation of Healthcare-Related Parameters

The errors corrupting the estimates of the *x*-*y* coordinates can significantly decrease the accuracy of the estimates of the healthcare-related parameters. To reduce the effect of the propagation of the errors of position estimation, the sequences of the *x*-*y* coordinates have been smoothed by means of a method based on a moving average filter.

The following indicators of uncertainty of estimation have been used:The mean error determined with respect to the corresponding reference value (ME);The standard deviation of that error (SE).

## 5. Results of Experimentation

### 5.1. Uncertainty of Estimation of Monitored Person’s Position

The estimates of the two-dimensional movement trajectories obtained in experiment EXP#1 are shown in Figure 5; the corresponding empirical distribution functions are provided in Figure 6. The values of the uncertainty indicators described in Section 4.2.1 are presented in Table 1. The analogous set of results, but obtained for experiment EXP#2, is provided in Figure 7 and Figure 8, and Table 2.

The analysis of the presented results is leading to the following conclusions:

The fused estimates of the person’s position are characterised by lower bias and dispersion when compared with the radar-data-based and depth-data-based estimates; moreover, the fused estimates are only slightly affected by the obstacles occluding a person. It should be stressed that in the case of the methods based on the artificial neural networks, the significant increase in the accuracy of the position estimation has been achieved despite the fact that the artificial neural networks were trained only on the synthetic data.In experiment EXP#1, where the monitored person was occluded by the obstacles, the more accurate estimates of the trajectories have been obtained by means of the NARX method than by means of the MLP method. This result can be explained by the fact that in the case of the recurrent neural network, the prediction of the person’s position at each time instant is based on the past position of that person. When the monitored person “disappears” behind the obstacle, larger measurement errors may corrupt the radar data and the depth data, but the recurrent neural network may react to such sudden changes and mitigate their influence on the result of the data fusion.If the values of the uncertainty indicators, calculated on the basis of the fused data, are concerned, the best overall results have been obtained for the NARX method: it is reflected in the lowest values of the mean error and the median error, as well as the $A_{\mathrm{ECDF}}$ indicator. Even though the values of the maximum error and the standard deviation of the errors are slightly greater when compared to the other methods, it does not affect the overall performance of the NARX method.

While the differences among the values of the $A_{\mathrm{ECDF}}$ indicator may seem insignificant, it is worth noting that the decrease in its value by only a few hundredths may indicate a quite significant distortion of the estimated trajectories. In Figure 9, the estimates of the movement trajectories from EXP#1, obtained on the basis of the data fused by means of the NARX method and by means of the WINFNS method, minimisation of weighted infinity-norm [37], are presented; for the NARX method (Figure 9a), the value of the $A_{\mathrm{ECDF}}$ indicator is 0.85, while for the WINFNS method (Figure 9b), it is 0.83. Moreover, even a small reduction of the uncertainty of the position estimates may have significant impact when those estimates are used for determining quantities carrying information important for medical and healthcare services.

Finally, during the experiments, it has been observed that the number of neurons in the hidden layer varying from 6 to 10 does not influence significantly the results of the data fusion; however, in the case of a nonlinear autoregressive network with exogenous inputs, the increase in the number of the hidden neurons may make the training more challenging because more complex networks are prone to overfitting.

### 5.2. Uncertainty of Estimation of Healthcare-Related Parameters

#### 5.2.1. Experiment EXP#1

The values of the uncertainty indicators, obtained on the basis of the radar data, depth data, and fused data, acquired in EXP#1, are presented in Table 3.

If the EXP#1 results, obtained for the radar data and for the depth data, are considered, it can be noticed that:In the case of the radar data, the number of turns is significantly underestimated. This may be explained by the smoothing of the radar data during their preprocessing: as a result, two consecutive turns are often treated as one.In the case of the depth data, the travelled distance is significantly underestimated. This may be explained by the obstacles occluding the monitored person and making the tracking impossible.The radar-data-based and depth-data-based estimates of the mean walking speed are similarly accurate.

If the EXP#1 results obtained for the fused data are considered, it can be noticed that:The values of the uncertainty indicators are generally lower than the uncertainty indicators obtained for the radar data and for the depth data.The estimates of the number of turns, obtained for the MLP method and the NARX method, are considerably more accurate than those estimates obtained for the KF method—in the case of the last method, the number of turns for each realisation of the scenario has been overestimated by almost one. This may be explained by the corruption of the depth data related to “disappearing” of the monitored person behind the obstacles and the inability of the KF method to mitigate this phenomenon.

#### 5.2.2. Experiment EXP#2

The dependence of the values of the uncertainty indicators on the walking speed is presented in Figure 10, Figure 11 and Figure 12. The analysis of those results leads to the following conclusions:In the case of the estimates of the number of turns, the values of the ME indicator, determined on the basis of the radar data, are generally lower than zero—meaning that, regardless of the walking speed, the number of turns is underestimated; on the other hand, the values of the ME indicator, determined on the basis of the data fused by the KF method, are greater than zero, which means that the number of turns is overestimated. The best results are obtained for the depth data and for the data fused by means of the MLP method and the NARX method—the values of the ME indicator, determined on the basis of those data, are not significantly affected by the walking speed of the person.In the case of the estimates of the travelled distance and the estimates of the walking speed, the values of the ME indicator, determined on the basis of the data fused by means of the KF method, are slightly better than the values of this indicator, determined on the basis of the data fused by means of the other methods. Moreover, for all the methods of data fusion, the values of the ME indicator decrease with the increase in the walking speed of the monitored person. This phenomenon is caused by the deviations of the estimates of the walking trajectory around the corners of that trajectory—the greater the walking speed, the smoother the trajectory becomes and the smaller the estimates of the distance and walking speed.

## 6. Conclusions

In this paper, the useability of feedforward and recurrent neural networks for fusion of data from impulse-radar sensors and depth sensors, in the context of healthcare-oriented monitoring of elderly persons, has been investigated. A new method for fusion of data has been proposed, viz., a method based on a nonlinear autoregressive network with exogenous inputs (NARX method), and compared with a method based on a multi-layer perceptron (MLP method) as well as one reference method based on a Kalman filter (KF method). All the artificial neural networks have been trained on synthetic data. An extensive programme of experimentation, designed for comparison of all the methods, has involved localisation of a person within a monitored area, and the estimation of several parameters of gait, considered important for medical experts, viz., the number of turns made during walking, the travelled distance, and the mean walking speed.

The results of experimentation have confirmed that the artificial neural networks trained on synthetic data may be effectively used for fusing the data from the impulse-radar sensors and from the depth sensors. The best overall results have been obtained for the NARX method: in the case of the recurrent neural network, the estimate of the position of a person is based on the estimates of that position at previous time instants; therefore, such network may effectively react to sudden changes in the measurement data, caused by stops and occlusions, and mitigate their influence on the estimated movement trajectory. This cannot be achieved for the feedforward neural network without a feedback loop because it treats the measurement data independently, without taking into account the movement history.

## Figures and Tables

**Figure 1 sensors-23-01457-f001:**
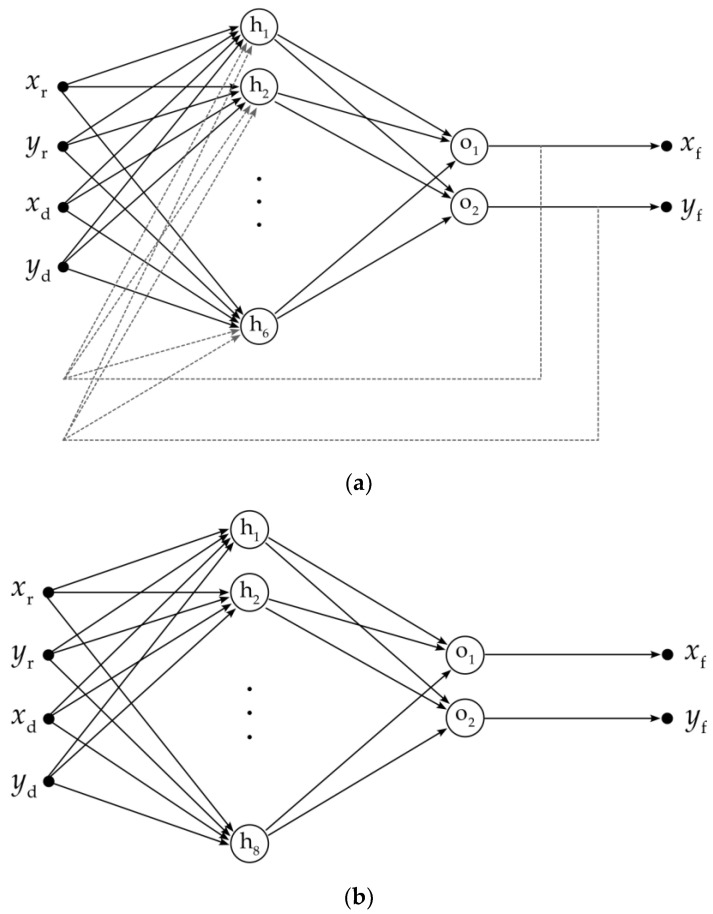
Artificial neural networks used for fusion of data: the nonlinear autoregressive network with exogenous inputs (**a**) and the multilayer perceptron (**b**); $x_{r}$ and $y_{r}$ denote the coordinates acquired by means of the impulse-radar sensor; $x_{d}$ and $y_{d}$ denote the coordinates acquired by means of the depth sensor; $x_{f}$ and $y_{f}$ denote the fused coordinates; $h_{1},\;\ldots ,\;h_{8}$ denote the neurons in the hidden layers; $o_{1}$ and $o_{2}$ denote the neurons in the output layers.

**Figure 2 sensors-23-01457-f002:**
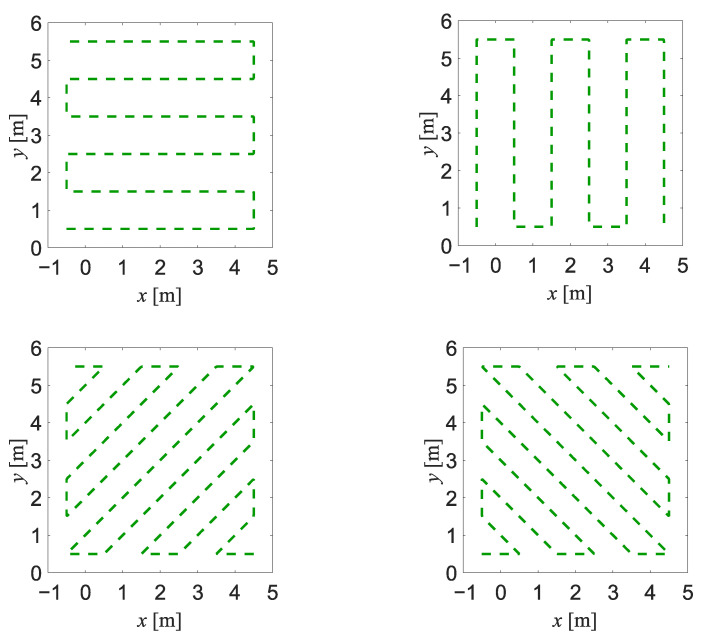
Reference trajectories used for generation of the synthetic data.

**Figure 3 sensors-23-01457-f003:**
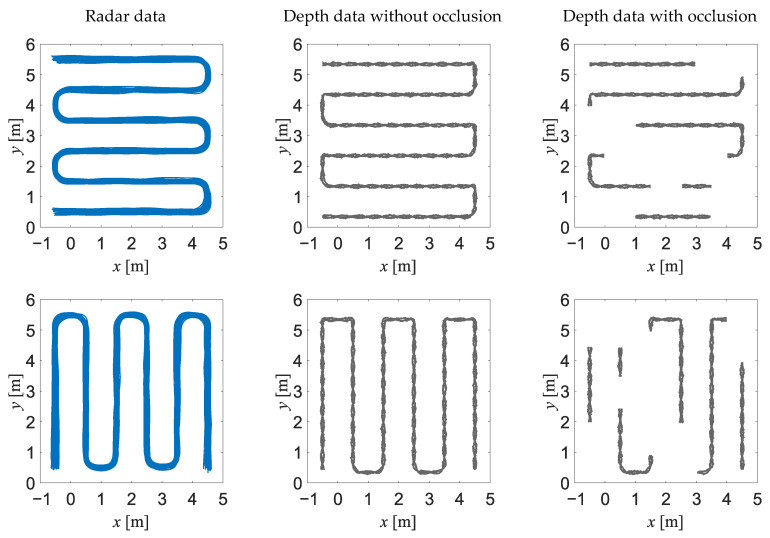
Synthetic data used for training the neural networks: the radar data (**left column**), the depth data without occlusion (**middle column**) and the depth data with occlusion (**right column**). Each graph for radar data depicts 40 superimposed sequences of synthetic data, while each graph for depth data depicts 20 superimposed sequences of synthetic data.

**Figure 4 sensors-23-01457-f004:**
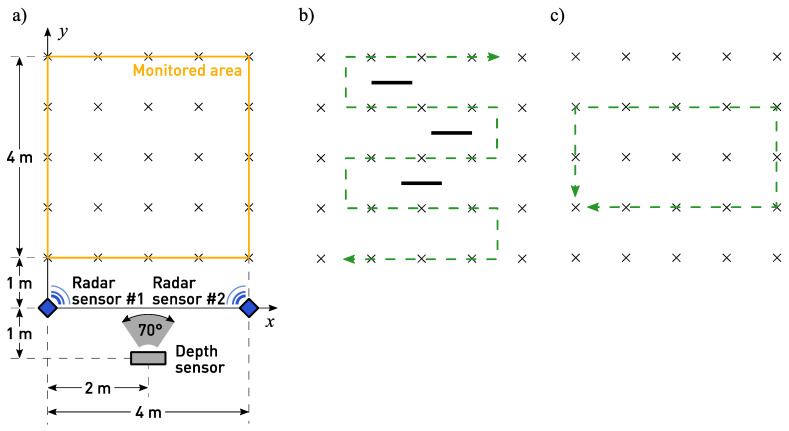
Experimental setup (**a**) and the movement scenarios considered in the experiments: the serpentine trajectory used in EXP#1 (**b**), and the rectangle-shaped trajectory used in EXP#2 (**c**). The reference points, i.e., the points where marks have been placed on the floor, are indicated with the crosses.

**Figure 5 sensors-23-01457-f005:**
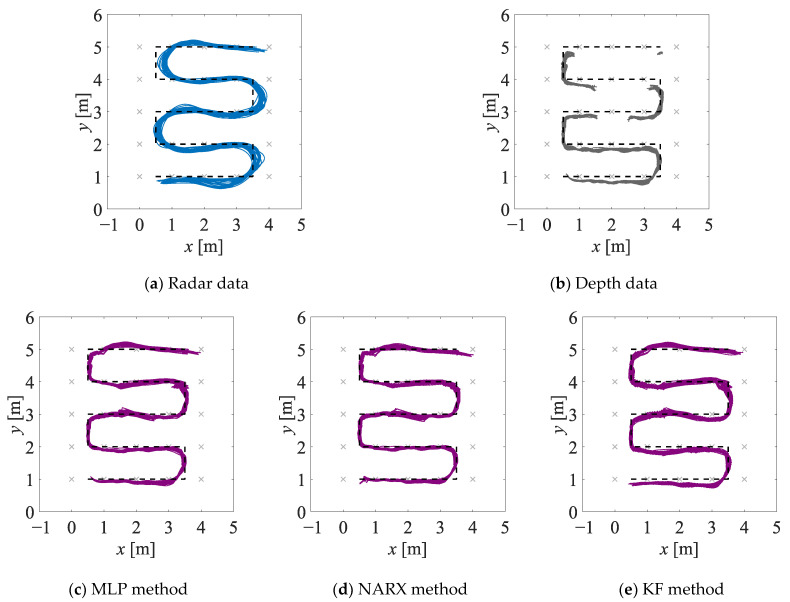
Estimates of the trajectories obtained in experiment EXP#1, on the basis of the radar data (**a**), the depth data (**b**), and the fused data (**c**–**e**); the black dashed lines denote the reference trajectories.

**Figure 6 sensors-23-01457-f006:**
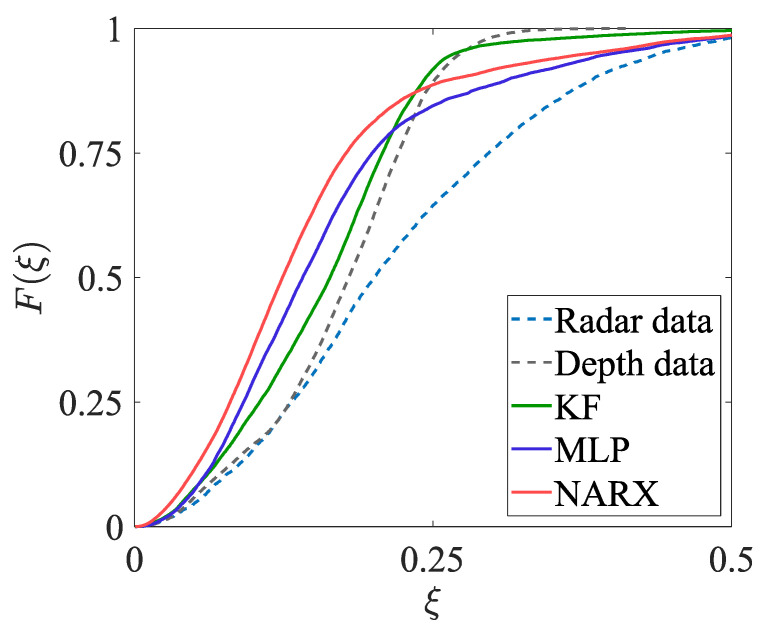
Zoom on the empirical cumulative distribution functions characterising the position errors, obtained in EXP#1.

**Figure 7 sensors-23-01457-f007:**
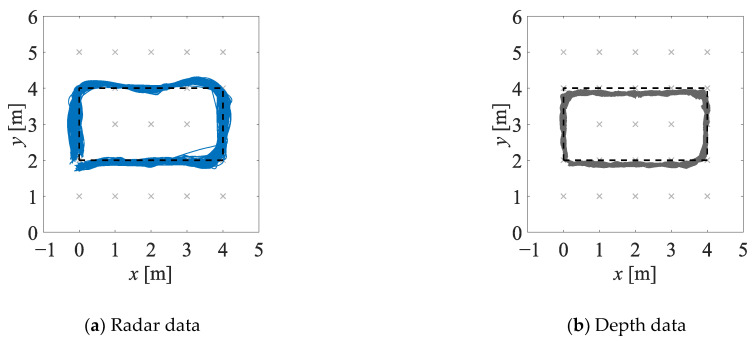
Estimates of the trajectories obtained in experiment EXP#2, on the basis of the radar data (**a**), the depth data (**b**), and the fused data (**c**–**e**); the black dashed lines denote the reference trajectories.

**Figure 8 sensors-23-01457-f008:**
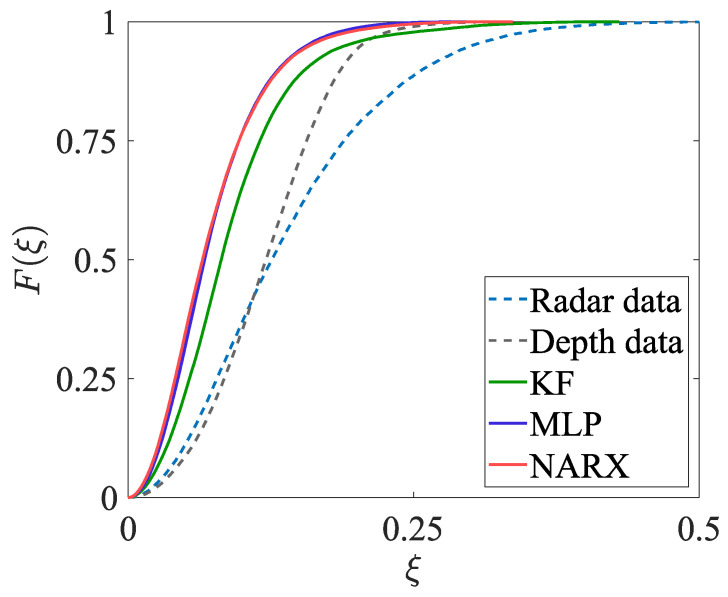
Zoom on the empirical cumulative distribution functions characterising the position errors, obtained in EXP#2.

**Figure 9 sensors-23-01457-f009:**
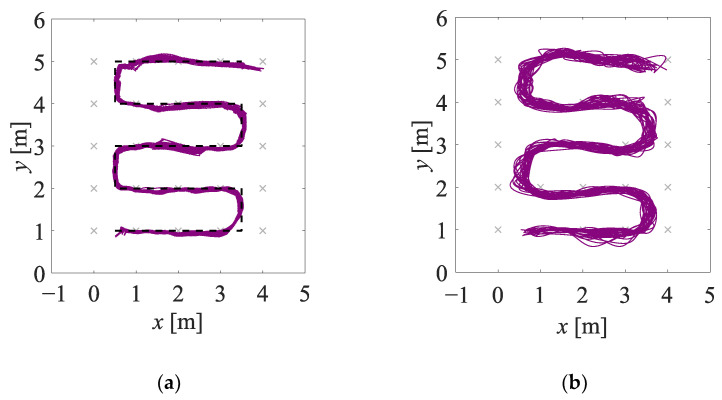
Estimates of the trajectories obtained in EXP#1, by means of the NARX method (**a**), and the WINFNS method described in [37] (**b**); for the NARX method, the value of the $A_{\mathrm{ECDF}}$ indicator is 0.85, while for the WINFNS method, it is 0.83.

**Figure 10 sensors-23-01457-f010:**
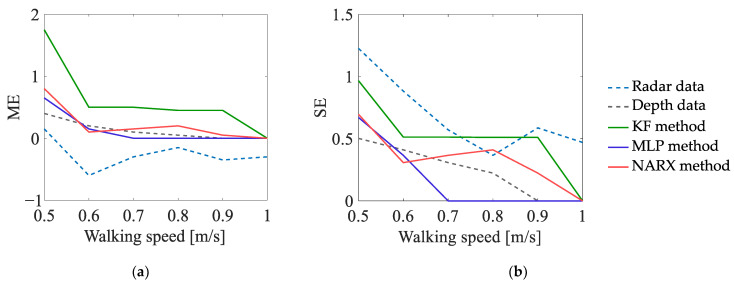
Uncertainty indicators characterising the estimates of the numbers of turns, obtained in EXP#2.

**Figure 11 sensors-23-01457-f011:**
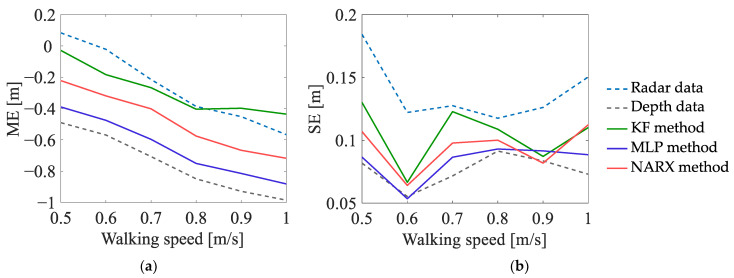
Uncertainty indicators characterising the estimates of the travelled distance, obtained in EXP#2.

**Figure 12 sensors-23-01457-f012:**
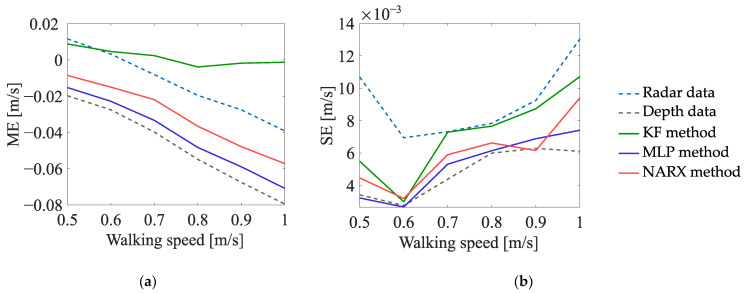
Uncertainty indicators characterising the estimates of the walking speed, obtained in EXP#2.

**Table 1 sensors-23-01457-t001:** Values of the uncertainty indicators, calculated on the basis of the position errors, obtained in EXP#1.

Uncertainty Indicator	Radar Data	Depth Data	MLP	NARX	KF
MEAE [m]	0.22	0.17	0.16	0.15	0.16
MEDE [m]	0.20	0.18	0.14	0.12	0.16
MAXE [m]	0.63	0.41	0.63	0.67	0.70
STDE [m]	0.12	0.07	0.11	0.10	0.08
$A_{\mathrm{ECDF}}$	0.78	0.83	0.84	0.85	0.84

**Table 2 sensors-23-01457-t002:** Values of the uncertainty indicators, calculated on the basis of the position errors, obtained in EXP#2.

Uncertainty Indicator	Radar Data	Depth Data	MLP	NARX	KF
MEAE [m]	0.14	0.12	0.07	0.07	0.09
MEDE [m]	0.13	0.12	0.07	0.07	0.08
MAXE [m]	0.60	0.33	0.32	0.34	0.43
STDE [m]	0.08	0.05	0.04	0.05	0.06
$A_{\mathrm{ECDF}}$	0.86	0.88	0.93	0.93	0.91

**Table 3 sensors-23-01457-t003:** Values of the uncertainty indicators obtained in EXP#1.

Uncertainty Indicator	Radar Data	Depth Data	MLP	NARX	KF
**Number of turns**					
ME	–6.00	–0.73	–0.13	0.07	0.93
SE	1.08	0.91	1.36	1.34	1.20
**Travelled distance [m]**					
ME	–1.43	–6.86	–0.74	–0.79	–0.53
SE	0.42	0.13	0.30	0.33	0.25
**Walking speed [m/s]**					
ME	–0.03	–0.04	–0.02	–0.02	–0.01
SE	0.01	0.00	0.01	0.01	0.01

## Data Availability

The data presented in this study are available on request from the corresponding author.

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
