# Peer review of "Application of Feedforward and Recurrent Neural Networks for Fusion of Data from Radar and Depth Sensors Applied for Healthcare-Oriented Characterisation of Persons’ Gait"

_sensors, 2023, doi:10.3390/s23031457_

Round 1

Reviewer 1 Report

No comments and suggestions.

Author Response

I would like to thank the Reviewer for providing a positive feedback.

Reviewer 2 Report

Pls see attached file.

Author Response

I would like to thank the Reviewer for providing both an overall positive feedback and constructive comments which have enabled me to enhance the manuscript. Please see the attachment for my detailed response. All the introduced changes have been highlighted in the revised version of the manuscript.

Reviewer 3 Report

    This paper proposes NARX, which applies the ANN trained with synthetic data, to fuse the measurement data from the impulse-radar sensors and depth sensors. For the localisation of a person walking and characterisation of persons’ gait, this paper provides an good insight.

Weaknesses:

(1). Some typos are supposed to be avoided. E.g. I am very confused that too many "-" appear in abstract.

(2). The structure of ANN and RNN mentioned in this paper should be shown in Figures.

Author Response

I would like to thank the Reviewer for providing both an overall positive feedback and constructive comments which have enabled me to enhance the manuscript. Please see the attachement for my detailed response. All the introduced changes have been highlighted in the revised version of the manuscript.
